# Characterization and Assembly Dynamics of the Microbiome Associated with Swine Anaerobic Lagoon Manure Treated with Biochar

**DOI:** 10.3390/microorganisms13040758

**Published:** 2025-03-27

**Authors:** A. Nathan Frazier, William Willis, Heather Robbe, Anna Ortiz, Jacek A. Koziel

**Affiliations:** 1United States Department of Agriculture-Agricultural Research Service (USDA-ARS), Bushland, TX 79012, USA; will.willis@usda.gov (W.W.); heather.robbe@usda.gov (H.R.); anna.ortiz@usda.gov (A.O.); 2Department of Agricultural and Biosystems Engineering, Iowa State University, Ames, IA 50011, USA

**Keywords:** ammonia, biochar, greenhouse gas emissions, microbial ecology, swine manure

## Abstract

Biochar has significant potential for livestock microbiomes and crop agriculture regarding greenhouse gas emissions reduction. Therefore, a pilot study was designed to investigate the effect of biochar application on the surface of swine manure from an open lagoon and the associated microbial communities. Samples were collected from four different treatment groups: control (*n* = 4), coarse biochar (*n* = 4), fine biochar (*n* = 4), and ultra-fine biochar (*n* = 4). Additionally, aged manure in bulk was collected (*n* = 4) to assess alterations from the control group. The method of 16S rRNA amplicon sequencing along with microbial analyses was performed. Diversity was significantly different between aged manure in bulk samples and all treatment groups (Kruskal–Wallis; *p* < 0.05). Additionally, distinct community compositions were seen using both weighted and unweighted UniFrac distance matrices (PERMANOVA; *p* < 0.01). Differential abundance analysis revealed four distinct features within all treatment groups that were enriched (q < 0.001): *Idiomarina* spp., *Geovibrio thiophilus*, *Parapusillimonas granuli*, and an uncultured Gammaproteobacteria species. Similarly, *Comamonas* spp. and *Brumimicrobium aurantiacum* (q-value < 0.001) were significantly depleted by all the treatments. Stochastic and functional analyses revealed that biochar treatments were not deterministically altering assembly patterns, and functional redundancy was evident regardless of compositional shifts.

## 1. Introduction

Commercial swine operations are of significant interest both to the global food market and in terms of the global concern of their environmental impact. China, the European Union (EU), and the United States are the global leaders in pork production, with China representing 50% of global pork production [1]. Commercial swine operations have shifted towards more intensive farming production to meet the growing demand for protein products. The global population has seen a remarkable three-fold increase across the 20th century, and as such, intensive swine farming systems have accounted for roughly 60% of global pork production [2,3]. Although these strategies have increased the availability of protein sources, intensive swine farming systems account for 35% of global greenhouse gas (GHG) emissions [4]. Indeed, the livestock sector overall has come under scrutiny in recent years for its contribution to climate change. It is estimated that anthropogenic emissions from livestock operations contribute approximately 31% of GHG emissions, 53% of methane (CH_4_) emissions, and nearly 80% of nitrous oxide (N_2_O) emissions [3,5]. While there are both theoretical and empirical studies ongoing for GHG mitigation [6], the livestock sector must meet the food demand while attempting to address these important environmental concerns.

Within crop and swine production systems, recent evidence suggests that feed production and feeding are the primary sources of GHGs [3]. Soybean meal is a primary protein source in pig diets, and consequently, soybean production is a significant contributor to GHG emissions. Interestingly, the same study labeled swine manure management as the second largest contributor to GHGs [3]. Within manure management, anaerobic lagoons are commonly used for the storage and treatment of livestock manures. These treatment lagoons are used to enhance the organic-rich materials within the manure and store the effluent. Treated manure is often used as a primary source of crop and pasture nutrients through land application [7,8]. Anaerobic lagoons rely on a consortium of microbial partners to degrade the organic material into various gases, including carbon dioxide (CO_2_), ammonia (NH_3_), and CH_4_. The microbiome of the lagoons is composed of anaerobic bacteria, facultative bacteria, aerobic bacteria, and archaeal methanogens. However, these microbial processes inevitably produce residual detrital material sludge. When accumulated, sludge can detrimentally affect lagoon treatment effectiveness and increase GHG emissions as well as odorous emissions [7,9]. Thus, it is crucial to identify sludge treatment options that mitigate GHG emissions. Furthermore, it is important to understand the microbial ecologies within commercial swine anaerobic lagoons to potentially identify key traits within the microbiome leading to these gaseous and odorous emissions.

Interestingly, biochar has the potential to mitigate NH_3_ and GHG when applied to manure storage systems. Biochar is high in carbon and is produced by using various biomass in processes such as pyrolysis, torrefaction, or gasification and has been shown to have many potential uses in crop and livestock agriculture [10], such as reducing CH_4_, CO_2_, and N_2_O emissions from manure [11]. Biochar is a cheaper alternative to other anaerobic digestion (AD) amendments and is known for its stable physiochemical properties [12]. Additionally, biochar amendments have been utilized in other agricultural systems to enhance pig manure composting capabilities [13], improve the AD of poultry manure [14], improve soil health parameters [15,16,17], and reduce gaseous and odorous emissions [18,19]. A review of biochar’s potential to mitigate gaseous emissions from stored manure is presented by Chen et al. (2024) [20].

Importantly, the roles of microbes in these processes are inherent. Microbial succession is tied to a series of organic matter decomposition stages that alter microbial functionality to dominate nutrient transformation [21]. While many studies exist to understand the role of biochar in composting processes [22], there is a lack of knowledge on how biochar can alter microbial ecologies in commercial swine anaerobic lagoon systems. Previous studies have characterized microbial species present in swine lagoons, and results have shown differences in microbial communities based on lagoon type (i.e., covered versus uncovered) [23,24]. However, little has been elucidated on how biochar treatments might alter microbial ecology and function.

Microbial ecology within anaerobic lagoons is key to effective manure storage and practical land-use techniques, as previously mentioned. Thus, it is important to understand both the key microbes involved in biochar-amended anaerobic lagoons and to understand the forces involved in microbial assembly. In the last decade, there have been great strides made in next-generation sequencing, allowing researchers to identify key microbial characterization and assembly features in various biological environments [25]. While it is assumed that deterministic forces such as lagoon treatments can alter microbial assemblies in a predictive manner, it remains unclear whether true environmental forces are correlated with microbial assemblages or whether stochastic forces play a greater role. Previous research indicated that microbial compositions were correlated with odor profiles in swine manure lagoon effluent, although this effect could not be differentiated from environmental factors [9]. Additionally, hydrogenotrophic methanogens were enriched within effluent from covered swine anaerobic lagoons and sludge, suggesting that hydrogenotrophic methanogenesis predominated in these systems [26]. However, the forces shaping these microbial communities and functional outputs have yet to be explained.

Therefore, the goal of this pilot study was to characterize the microbial communities of biochar-amended swine manure and analyze the relationship between these communities to GHG mitigation. We utilized an in vitro system and 16S rRNA amplicon sequencing to address three primary objectives: (i) determine which biochar treatments are associated with GHG mitigation; (ii) describe, if any, the microbial ecological differences arising from various biochar amendments; and (iii) describe, if any, the changes in assembly patterns that can be seen and the forces driving these assemblies.

## 2. Materials and Methods

### 2.1. Manure Collection, Biochar Selection, and Experimental Design

Swine manure was collected from a single aerobic open lagoon at a commercial swine operation in Oklahoma, USA, during the spring of 2024. Manure was collected in large, sealed buckets from various locations from the lagoon and transported back to the lab for experimental setup (USDA-ARS, Livestock Nutrient Management Research Unit, Bushland, TX, USA). While sampling occurred at only one location, single-location studies are valuable for controlling bias investigations of intervention details. Moreover, the manure was collected from an integrated swine finisher operation. Management of the farms involved in the operation is nearly identical, including facility design, manure management, breeding and genetics of the animals, feeding practices and feed rations, and veterinary interventions. Thus, variation in manure microbiomes was relatively controlled, reducing bias within the current study.

Retail-quality red oak charcoal was used as biochar with 16 open-storage containers used to simulate swine manure anaerobic lagoons. The three experimental biochar treatments were determined based on biochar particle size determined using pre-ground red oak charcoal that would pass through 5–10 (coarse), 10–50 (fine), and 50–100 (ultra-fine) mesh particle size sieves. An in vitro 4 × 4 completely randomized design was conducted with four treatment groups: control (CON), coarse biochar (COR-biochar), fine biochar (FIN-biochar), and ultra-fine biochar (UF-biochar). Additionally, there were four replicates per treatment. Biochar treatments were sprinkled on top of the manure to form a ~6 mm layer, as described by previous research [27,28], and pH, water absorption, and NH_3_, CH_4_, and CO_2_ emissions were recorded for 14 days. Biochar chemistry and temporal emissions reductions via biochar application were outside the scope of the current study and are part of a larger longitudinal, repeated-measures study (unpublished data). Therefore, only day 14 emissions data were used for microbial testing.

### 2.2. Microbial Sampling and DNA Extraction

At day 14, microbial samples were taken from each of the treatments. In total, treatment (*n* = 16 with four samples from each treatment replicate), pooled (*n* = 4), and aged manure iThin bulk (AMB; *n* = 4) liquid manure samples were collected in 1.5 mL microcentrifuge tubes. The samples were stored at −4 °C overnight, awaiting DNA extraction. The following day, DNA was extracted from all samples using the DNeasy PowerSoil^®^ Pro Kit (Qiagen, Valencia, CA, USA) following the manufacturer’s protocols. Extracted DNA samples were then stored at −80 °C until shipment for sequencing. Amplicon sequencing was conducted at a commercial laboratory (Molecular Research DNA, Shallow Water, TX, USA) using bTEFAP^®^ technology [29] and following the Earth Microbiome Project protocol (www.earthmicrobiome.org, accessed on 1 June 2024). Briefly, 96-well plates were prepped with the extracted DNA, a negative extraction control, and a mock community positive control. Sequences were placed in the wells at random to avoid confounding due to technical artifacts. The extracted DNA was amplified using 515f/806r primers targeting the variable V4 region of the 16S rRNA subunit [30,31]. The HotStarTaq Plus Master Mix Kit (Qiagen, Valencia, CA, USA) was used in a 30-cycle PCR under the following conditions: 95 °C for 5 min, followed by 30 cycles of 95 °C for 30 s, 53 °C for 40 s, and 72 °C for 1 min. A final elongation step was performed at 72 °C for 10 min. The final PCR products were checked for success using a 2% agarose gel based on the relative intensity of the gel bands. Samples were multiplexed using unique indices and pooled together in equimolar concentrations used for sequencing. Pooled samples were purified using calibrated Ampure XP beads (Beckman Coulter, Indianapolis, IN, USA). Purified multiplexed samples were then sequenced using the Illumina MiSeq platform (Illumina, Inc., San Diego, CA, USA).

### 2.3. Sequencing and Microbial Ecology Analysis

Following sequencing, multiplexed sequences were joined, primers were removed, and sequences < 150 base pairs (bp) and ambiguous base calls were removed. Sequences were then quality-filtered for chimeras and demultiplexed using a maximum expected error threshold of 1.0. Then, the demultiplexed sequences were analyzed further by our research group following similarly published methods using Quantitative Insights into Microbial Ecology (QIIME2) version 2023.7 [25,32]. QIIME2 was accessed using the United States Department of Agriculture-Agricultural Research Service’s SCINet Scientific Computing Ceres high-performance computing cluster (ARS project numbers 0201-88888-003-00D and 021-88888-002-000D). Demultiplexed sequences were denoised using the QIIME2 dada2 plugin, generating amplicon sequence variants (ASVs) [33,34]. Sequences were rarefied at a sampling depth of 60,000 ASVs/sample based on alpha rarefaction curves and visualizing the sampling depth within QIIME2 to filter low-quality reads based on the Phred quality score. Taxonomic classification was performed utilizing the SILVA 138 99% database from the QIIME2 feature-classifier plugin [35,36]. ASVs assigned as mitochondria or chloroplasts were then filtered from the dataset. A phylogenetic insertion tree was constructed via the SEPP algorithm within the QIIME2 fragment-insertion plugin using the SILVA 128 tree as a reference backbone [34,37]. The constructed phylogenetic tree was used in downstream microbial ecology analyses.

To begin with, sequences were analyzed for compositional, diversity, and taxonomic differences. Phylogenetic compositional and diversity analyses were executed using the QIIME2 core-metrics pipeline [32,36,37]. Beta diversity was tested statistically via permutational multivariant analysis of variance (PERMANOVA) with multiple testing corrections using the QIIME2 pairwise plugin and was tested over 999 permutations to control Type I error. Beta diversity was used to analyze compositional differences between the treatment groups using both unweighted and weighted UniFrac distances [38,39]. Unweighted UniFrac distance measures the phylogenetic distance between communities, reflecting differences in community membership based on the presence or absence of taxa. Weighted UniFrac distance incorporates both phylogenetic relationships and the abundance of taxa, reflecting differences in community composition changes emphasizing changes in dominant taxa. Additionally, the QIIME2 permdisp plugin was used for multivariate dispersion measurements to rule out compositional differences due to high dispersion within each treatment [40]. Composition was tested against pH using Bray–Curtis distance with PERMANOVA using the adonis2 function within the “vegan” package in R (R version 4.4.2) in RStudio version 2024.12.0 [41,42,43]. PERMANOVA adonis2 used 999 permutations to mitigate Type I error by creating a null distribution. Significant results (*p*-value < 0.01) indicate that pH was correlated with microbial community composition.

Alpha diversity was analyzed using the observed richness, Shannon’s index, and Faith’s phylogenetic diversity (PD) metrics [44,45,46]. Briefly, observed richness is a count of the total unique ASVs present in the dataset, Shannon’s index is a measurement of both observed richness and evenness (distribution of species abundance), and Faith’s PD measures observed richness while accounting for phylogenetic relationships [25]. Alpha diversity measurements were tested statistically using a Kruskal–Wallis pairwise test with a Benjamini–Hochberg multiple testing correction, which computes both a *p*-value and an FDR-adjusted *p*-value (q-value) [47]. Alpha and beta diversity were tested at alpha level = 0.05. Differential abundance testing for taxonomic analysis was assessed using analysis of composition of microbiomes with bias corrections (ANCOM-BC) for each of the treatment groups [48]. ANCOM-BC measures differentially abundant taxa between treatment groups while accounting for the compositional nature of microbiome datasets. Further, ANCOM-BC inherently controls both Type I and Type II errors, producing a false discovery rate (FDR)-adjusted *p*-value (q-value). The top five enriched and depleted features were recorded at an alpha level = 0.001. The identified features were then taxonomically identified using the National Center for Biotechnology Information’s blastn suite [49,50].

Next, microbial community assembly was assessed using normalized stochasticity testing (NST) using the R package “NST” following previously published methods [25,51,52]. Due to NST’s inherent reliance on abundance-based comparisons, Bray–Curtis distance was used for NST testing, as it captures abundance changes better than other beta diversity metrics [42]. NST testing using the Bray–Curtis distance matrix calculated the modified Raup–Crick distance (β_RC_), the standardized effect size index (SES), the normalized stochasticity index (NSTi), and the modified stochasticity ratio (MST) [52,53,54,55]. Briefly, β_RC_ values approaching zero indicate stochastic assembly governed by neutrality. Similarly, SES values close to zero indicate neutrality, while values > 2 and <−2 indicate deterministic assembly. NSTi values greater than 0.5 indicate stochasticity, with values approaching 1 indicating strong stochastic assembly. MST values produce a percentage of >50%, favoring stochasticity [25]. Nonmetric multidimensional scaling (NMDS) plots were used to display community ordination based on β_RC_ values. NST testing metrics were tested statistically using Dunn’s test with Benjamini–Hochberg multiple testing correction found in the “FSA” R package [56,57]. Significance was set at alpha level = 0.01 to further reduce FDR and to ensure robust, biologically meaningful differences were captured. Failing to reject the null hypothesis (*p* > 0.01) indicates that the assembly metrics are statistically true and their corresponding indications are valid.

Next, we evaluated whether there were differences in microbial taxa that were driving community composition between the treatment groups. Similar to Frazier et al. [25], we used the “DirichletMultinomial” package in R to generate a Dirichlet multinomial model (DMM), which uses the lowest Laplace approximation score to cluster samples based on microbial community structure similarity [58,59]. The top five microbial families with the highest DMM approximation scores were recorded. Clusters were analyzed using Monte Carlo simulation permutation testing to determine the significance of DMM-produced clusters using the “parallel” package in R [41,60]. For example, if clustering produces only one cluster in our dataset, Monte Carlo simulations generate randomized datasets to determine whether these datasets produce more than one cluster. If the simulated data produce multiple clusters, then the lack of clustering in the real dataset is statistically meaningful. Thus, the identified taxa at the family level can be considered primary drivers in community assembly.

Finally, the functional capabilities of the microbial communities were assessed using phylogenetic investigation of communities by reconstruction of unobserved states (PICRUSt2) [61]. PICRUSt2 infers gene content from 16S rRNA amplicon data based on the phylogenetic relationships with the reference genome. Functional pathways were predicted using the KEGG database [62], and relative abundances were normalized using a z-score. Predicted functions were mapped to KEGG orthologs (KOs) and metabolic pathways. Kruskal–Wallis with FDR-adjusted testing was used to identify any statistically different KOs across the treatment groups (FDR-adjusted *p*-value). All visualizations were generated using the “ggplot2” package in R [63].

### 2.4. Statistical Analysis

Gaseous emissions flux percentage reduction (%R) was calculated by the following equation:(1)$$\%R=\frac{C_{\mathrm{control}}-T_{\mathrm{biochar}}}{C_{\mathrm{control}}}$$
where %R is the percentage of mitigation, C_control_ is the measured gas flux (mass/area/time) in the headspace of the untreated manure, and T_biochar_ is the measured gas flux in the headspace of the biochar-treated manure. Each treatment group was statistically analyzed in R using Kruskal–Wallis testing, as %R are typically non-normally distributed. If significant results were determined, Dunn’s post hoc tests with Benjamini–Hochberg multiple testing correction were conducted to identify the specific pairwise differences between treatments while controlling Type I error. The alpha level was set at 0.05 for gaseous emissions statistical analysis. Additionally, pH was statistically analyzed using a one-way analysis of variance (ANOVA) test, and statistically significant results were tested using Tukey’s honest significant difference (HSD) test for pairwise comparisons [64,65]. The alpha level was set at 0.001 for pH analysis to better control for FDR.

## 3. Results

### 3.1. Analysis of NH_3_, H_2_O, CH_4_, and CO_2_

Gaseous emissions were analyzed to determine whether there was a difference in flux percentage reduction between the treatment groups. The mean, median, and standard deviations for each gaseous flux percentage reduction are highlighted in Table 1, and raw flux percentage reductions and pH levels can be found in Appendix A. All treatments reduced NH_3_, although there were two COR-biochar replicates and one UF-biochar replicate that did not see a reduction. Similarly, all treatments reduced H_2_O, with one UF-biochar replicate not showing a percentage reduction. All treatments and all replicates saw a percentage reduction for CH_4_. No treatment group or replicate reduced CO_2_. Kruskal–Wallis tests confirmed that there were statistically significant differences between the treatments and CH_4_, H_2_O, and CO_2_ flux percentage reduction (Table 1; *p*-value < 0.05). There were no significant differences between the treatments and NH_3_. Dunn’s test was used to determine the specific differences between the treatments and flux percentage reduction for CH_4_, H_2_O, and CO_2_ (Appendix A). For CH_4_, FIN-biochar (*p* = 0.01, q = 0.04) and UF-biochar (*p* = 0.0007, q = 0.004) were significantly different from CON. The COR-biochar treatment was thought to be statistically different from UF-biochar (*p* = 0.04); however, when controlling FDR, it was determined that differences between the two treatments were not likely (q = 0.08). Similarly, COR-biochar (*p* = 0.02) and FIN-biochar (*p* = 0.01) were statistically different from CON for H_2_O flux percent reduction; however, controlling for FDR revealed that neither treatment was likely biologically different (q = 0.06 for both treatments). There was a statistical difference between UF-biochar and CON (*p* = 0.01, q = 0.04) for CO_2_ flux percentage reduction. Both FIN-biochar (*p* = 0.03, q = 0.06) and COR-biochar (*p* = 0.01, q = 0.06) were not likely biologically different. Statistical analysis of pH revealed strong statistical differences between CON and the treatment groups (*p*-value < 0.001). No differences were seen between the treatment groups.

### 3.2. Sequencing Results

A total of 24 samples were taken for microbial analysis. Paired-end 16S rRNA amplicon sequencing yielded a total of 19,480,613 sequencing reads, with 1,967,048 sequences per sample (mean frequency per sample = 81,960.3). Following denoising, quality filtering, chimera removal, and further filtering steps, we obtained a total of 3017 unique ASVs, with a mean feature frequency of 652 ASVs/sample. Quality-control checks on the negative control indicated that contamination of samples during DNA extraction was not likely. Analysis of the mock community (positive control) indicated that the sequencing run was valid. These quality-control checks indicated that it was reasonable to remove the controls from the dataset. Given this, downstream analysis could be utilized to answer the primary study questions.

### 3.3. Characterization of Biochar-Amended Swine Manure Microbiomes

The microbiomes of biochar-amended swine manure indicated distinct microbial compositions between the treatment groups. Both weighted and unweighted UniFrac distances revealed that microbial communities were distinct between the treatment groups and were visualized using principal coordinate analysis plots (PCoA; Figure 1). PERMANOVA analysis indicated that there were statistical differences in community membership (unweighted; *p* < 0.001, q < 0.001) and community composition (weighted; *p* < 0.01, q < 0.01). Multivariate dispersion analysis indicated that there were no significant differences between the treatments for both weighted (*p* = 0.696) and unweighted (*p* = 0.577), indicating that observed group differences were not due to within-group variability caused by dispersion. Thus, community differences using both distances were confirmed. Pairwise PERMANOVA analysis of weighted UniFrac distance indicated statistical differences between AMB and COR-biochar (*p* < 0.05), CON (*p* < 0.05), and FIN-biochar treatments (*p* < 0.05), as well as between FIN-biochar and UF-biochar (*p* < 0.05). FDR-adjusted q-values of q = 0.1 were reported for all pairwise interactions. For unweighted UniFrac distance, pairwise PERMANOVA analysis revealed statistical differences between AMB and COR-biochar (*p* < 0.05, q = 0.06), CON (*p* < 0.05, q = 0.06), and FIN-biochar (*p* < 0.05, q = 0.06). Additionally, differences were seen between COR-biochar and UF-biochar (*p* < 0.05, q = 0.06), as well as between FIN-biochar and CON (*p* < 0.05, q = 0.06) and between FIN-biochar and UF-biochar (*p* < 0.05, q = 0.06). Importantly, pH was a factor in shaping microbial composition, as pH from the top (*p*-value < 0.01, R^2^ = 0.17447) and the bottom (*p*-value < 0.001, R^2^ = 0.16743) of the biochar-treated manure system was statistically different. However, the R^2^ values indicate that pH only explained 17% of the variation between treatment groups.

Diversity differences were seen between the treatment groups (Figure 2A). For observed richness, AMB was statistically different for COR-biochar (*p* < 0.05, q = 0.1) and FIN-biochar (*p* < 0.05, q = 0.1), with richness being lower in AMB. Additionally, FIN-biochar had higher richness compared to UF-biochar (*p* < 0.05, q = 0.1). Shannon’s index revealed statistical differences for AMB and COR-biochar, CON, FIN-biochar, and UF-biochar (*p* < 0.05, q = 0.1). Trends could be identified for differences between UF-biochar and CON and FIN-biochar (*p* = 0.07, q = 0.1). Shannon’s index was highest for FIN-biochar and lowest for AMB (*p* = 0.03). No statistical differences were seen for any of the treatments when assessing Faith’s PD (*p* > 0.05). However, trends were seen for differences between AMB and CON, and FIN-biochar (*p* = 0.07, q = 0.1). Additionally, UF-biochar also saw trends in differences between CON and FIN-biochar (*p* = 0.07, q = 0.1).

Taxonomic profiles were assessed for each treatment type at the phylum level and the family level (i.e., relative abundance). The top bacterial phyla identified were Firmicutes A (28%), Bacteroidota (18%), Firmicutes D (15%), and Proteobacteria (14%; Figure 2B). The top archaeal phylum was Halobacteriota (7%). At the family level, *Clostridiaceae* (14%), *Bacteroidales UBA932* (8%), and *Acholeplasmataceae* (5%) were the top identified bacterial families (Appendix A). The top archaeal families reported were *Methanotrichaceae* (3%) and *Methanocorpusculaceae* (3%). Differential abundance testing produced the top five enriched and depleted taxa for each treatment group (Figure 3). Across all treatments and CON, *Idiomarina* spp. and *Parapusillimonas granuli* were enriched (q < 0.001), while an uncultured *Comsmonsd* spp. and *Brumimicrobium aurantiacum* were depleted (q < 0.001). For treatment groups only, *Geovibrio thiophilus* and an uncultured *Gammaproteobacteria* were enriched (q < 0.001). An uncultured archaeal species was depleted in both FIN-biochar and UF-biochar (q < 0.001). Curiously, a separate uncultured archaeal species was enriched in the FIN-biochar treatment (q < 0.001).

### 3.4. Characteristics of Microbial Assembly and Predicted Function

The microbial assembly patterns were assessed to characterize the mechanisms in which compositions were affected by treatments. Bray–Curtis distance revealed separation between the samples based on β_RC_ values (Appendix A). NST assessment revealed that all treatment groups had an NSTi value of >0.70, MST > 64%, SES ~0.5, and β_RC_ ~0.25 (Appendix A). The high NSTi and MST values indicate that stochastic processes predominantly shaped all treatment group microbial assemblages. Mild deterministic influences were indicated by SES values, with slightly stronger environmental filtering in AMB communities. However, the near-zero β_RC_ values confirmed random assembly, supporting neutral theory predictions. Kruskal–Wallis and Dunn’s statistical testing confirmed that the NST metrics were valid (*p*-value > 0.05). Additionally, DMM reported the top five microbial drivers of community assembly in each treatment group (Figure 4). The family *Methanotrichaceae* was the dominant driver in all treatment groups. Because distinct microbial clusters were not evident, Monte Carlo simulations were tested to confirm whether there was evidence of additional clustering. In all 100 simulation models, only one cluster was determined. This evidence indicates that there are no strong community shifts across treatments, and the lack of distinct clustering implies that stochastic processes and/or neutrality may be dominating assembly rather than deterministic treatment effects.

Predicted KOs were assessed using PICRUSt2, and it was revealed that the samples did not show distinct cluster patterns based on treatment type following Z-score normalization (Figure 5). Additionally, no statistically significant differences in KO abundance were observed across treatment groups using Kruskal–Wallis testing (FDR-adjusted *p*-value > 0.05). There were slight differential enrichments of various KOs, including acetaldehyde dehydrogenase, glucose-1-dehydrogenase, 1-deoxy-D-xylose-5-phosphate reductoisomerase, and aspartate-semialdehyde dehydrogenase. Additionally, sporadic KO enrichments were seen; however, they were not consistently linked to specific sample groups. PICRUSt2 also assessed predicted functional pathways, where Z-score normalization indicated differential and sporadic enrichment of specific pathways such as respiration, amino acid degradation, and amine and polyamine degradation; however, these patterns were not consistent enough to drive overall sample clustering (Appendix A). Kruskal–Wallis testing indicated no significant differences in pathway enrichment across the treatment groups (FDR-adjusted *p*-value > 0.05). Together, these observations indicate functional redundancy across the treatment groups, suggesting that diverse microbial taxa maintain similar functional capacities. As a result, ecosystem stability remains despite the compositional shifts seen in the dataset. Therefore, stochastic assembly is the underlying process contributing to functional stability, which promotes functional redundancy as the ecological outcome.

## 4. Discussion

Anaerobic lagoons are used in commercial swine operations to store and treat manure for future land-use applications. The essential microbial processes within these lagoons degrade organic material and concentrate the nutrients within the stored waste materials. However, gaseous and odorous emissions as by-products from these lagoons are cause for environmental and public concern. Indeed, these emissions have led to commercial swine operation closures and civil lawsuits [66]. Given the importance of anaerobic lagoons to swine operations, it is crucial to identify novel methods for GHG emissions mitigation and odor control. Thus, the current study investigated the application of biochar to pig manure in vitro to analyze GHG emissions and assess the importance of the microbiome across various biochar treatment types. Similar to previous research, NH_3_ and CH_4_ were reduced across treatment types; however, only CH_4_ was statistically significant for all treatments (*p*-value < 0.01) [11]. In contrast, CO_2_ was not successfully mitigated within the current study, and these findings were statistically significant (*p*-value < 0.05). As seen in previous studies, the high porosity of biochar could contribute to the failure to reduce CO_2_, and as such, smaller pores could produce better mitigating results [67]. The difference in the type of biochar could also alter its mitigation capacity. Dunn’s testing with FDR control revealed differences between CON, FIN-biochar, and UF-biochar in CH_4_ reduction (q-value < 0.05). Thus, finer biochar was more correlated with CH_4_ reduction, suggesting that the type of biochar used can contribute to GHG reduction [11]. Due to this, the microbial ecologies across the treatment types were evaluated to examine changes to the structure and predicted functionality that could be related to GHG emissions.

As seen in previous research, biochar-amended manures were separated from AMB (PERMANOVA, *p*-value < 0.01, q-value < 0.01) in terms of both membership and composition, suggesting that biochar amendments altered microbial ecologies from their original state (i.e., pig gut microbiome) [68]. Pairwise comparisons also reported compositional changes between the various treatment groups (*p*-value < 0.05), further indicating that biochar type plays a role in anaerobic lagoon treatment. Additionally, alpha diversity was statistically lower in AMB compared to the treatment groups (*p*-value < 0.05), highlighting that biochar amendments were creating an altered microbial state. However, as seen in other studies, there were no statistically significant differences between the treatments themselves [69]. These results suggest that biochar amendments altered manure composition and diversity from their original states, although treatment types may not have affected composition and diversity. Our results further suggest that the type of biochar used (i.e., fine or coarse) did not influence microbial diversity between treatment types. Therefore, because diversity was similar across treatment types, it can be inferred that there would be little variation in taxonomic identity.

To test this hypothesis, taxonomy was analyzed at the phylum and family level. The taxonomic analysis reported that the top bacterial phyla were Firmicutes, Bacteroidota, and Proteobacteria regardless of treatment type, demonstrating similarity to previously published results [69]. Interestingly, *Actinobacteria* has been considered a primary organic matter degrader in composting systems [70]. *Actinobacteria* was not among the top phyla in the current study, suggesting that different microbial partners are involved in anaerobic lagoons versus swine manure composting, regardless of original manure content. While this finding might not be surprising, it highlights that different manure management practices could rely on different microbial communities to achieve proper stored manure outcomes. At the family level, many of the top 30 families were previously reported as ubiquitous in anaerobic lagoon systems [23,24]. Within our study, these families include *Clostridiaceae*, *Peptostreptococcaceae*, *Synergistaceae*, and *Turicibacteraceae*. Of note, species that fall into the *Clostridiaceae* family have previously been reported in both swine manure biosolids and within lagoon slurries [71]. Moreover, our results support recent research highlighting that *Terrisporobacter*, belonging to the family *Peptostroptococcaceae*, and *Clostridium* sensu, belonging to the family *Clostridiaceae*, are key bacteria in fermentation processes in swine manure digestion [72,73].

Differential abundance testing revealed that *Idiomarina* spp. and *Parapusillimonas granuli* were enriched across all treatment types (q-value < 0.001). The enrichment of *Idiomarina* spp. contrasts with previous studies where *Idiomarina* was inhibited in treated manure compost [74]. *Idiomarina* spp. are known halophiles and have been linked to organic matter degradation, potentially suggesting a role in enhancing nutrient turnover [75]. The enrichment of these bacteria further suggests that biochar amendments could be shifting salinity or osmotic stress, giving way to altered microbial niches. Of notable interest, *Geovibrio thiophilus* was enriched in all treatment groups but was not enriched in CON (q-value < 0.001). *G. thiophilus* is a known sulfur-reducing bacterium and has been identified in sulfur control in anaerobically treated wastewater [76,77]. Although not directly tested in the current study, the enrichment of *G. thiophilus* suggests that sulfur might be reduced in biochar-amended swine manure. Importantly, sulfur-reducing bacteria can outcompete methanogens for H_2_ and acetate, inhibiting methane production [78]. Our study aligns with these findings, as biochar successfully reduced CH_4_ in all treatment types (*p*-value < 0.01). Thus, the enrichment of these taxa indicates the potential ability of biochar to alter the physiochemical environment of swine manure. The changes in the physiochemical environment could, therefore, result in microbial assembly and functional shifts that reduce GHG emissions. Therefore, microbial assembly dynamics and predicted functionalities were analyzed to corroborate these insights. Understanding how microbial communities assemble in swine manure lagoons could provide valuable insight into how GHGs or odorous amendments might be used.

Our study indicates that microbial communities in biochar-amended swine manure assembled in a stochastic, unified neutral theory-based manner based on NST metrics (*p*-value > 0.05). These results are in direct contrast with previously reported results that microbial assemblies in pig manure slurry were more niche-based than neutral [79]. Additionally, Peces et al. [80] reported that deterministic factors were dominant in anaerobic bioreactors using manure from a cattle-only slaughterhouse anaerobic lagoon. However, these discrepancies could be due to the addition of biochar to the swine manure. While the CON group showed that stochastic assembly forces were favored, pH was statistically different between treatment groups and CON. Previous studies have indicated that environmental factors such as pH can contribute to deterministic microbial assembly in swine manure [9,79]. Interestingly, our results indicate that pH was not statistically different between treatment types (*p*-value > 0.05), although pH was identified as a contributor to microbial composition variation (PERMANOVA *p*-value < 0.001). However, the model only explained 17% of the variation seen in community composition, suggesting that stochastic forces are likely to dominate in biochar-amended swine manures. To further support our indication of stochastic assembly, DMM models produced only one cluster and similar top bacterial drivers across all treatment groups, CON and AMB. Additionally, Monte Carlo simulations produced only one cluster across 100 simulations. These findings imply that deterministic selection pressures are not strong enough to create compositional divergence.

Our results further indicate that stochastic assembly as the top drivers are phylogenetically diverse. The dominance of stochastic processes may have been influenced by environmental homogeneity across treatments. Although pH explained a modest portion of compositional variation (~17%), the overall uniformity in key environmental parameters in our in vitro system could create conditions that limit strong niche differentiation. This hypothesis aligns with the neutral theory prediction that within homogenous environments, stochastic processes such as ecological drift and random dispersal dominate community assembly [81]. Importantly, the PICRUSt2 functional results indicated that samples were not distinctly clustered for either KOs or functional pathway predictions, suggesting that functional redundancy is an outcome of stochastic input. These findings indicate that functional stability might be achieved within these systems [5,82]. The observed functional redundancy, despite compositional shifts, aligns with the functional resilience concept. Functional resilience suggests that microbial communities maintain ecosystem functionality even with taxa turnover due to stochasticity [83]. Our results suggest that biochar-amended swine manure microbiomes possess a form of ecological insurance [84], where diverse microbial taxa with overlapping metabolic functions provide a buffer against environmental perturbations. This redundancy could explain why CH_4_ emissions were mitigated despite compositional shifts, suggesting that functionally similar taxa compensated for changes in microbial structure.

Importantly, Vellend et al. [85] provided a unification of niche and neutral theory by proposing that four key processes shape community assembly: selection (determinism), drift (stochasticity), dispersal (stochastic or deterministic), and speciation (less relevant in microbial ecology on shorter timescales). Herein, our results indicate that deterministic forces such as environmental filtering (i.e., pH) or niche modification could be acting on microbial assembly. Additionally, the repetition of microbial community drivers such as *Methanotrichaceae*, *Clostridiaceae*, *Peptostreptococcaceae*, and *Methanocorpusculaceae* across all groups could indicate deterministic selection based on functional traits. The functional group concept highlights these findings by grouping phylogenetically distinct organisms based on shared metabolic function [86]. Further still, the top driver across all groups was *Methanotrichaceae*. Indeed, *Methanotrichaceae* was once considered a strict acetoclastic methanogen until recent research determined that it could reduce CO_2_ to CH_4_ by accepting electrons from direct interspecies electron transfer (DIET) [87]. This metabolic plasticity challenges traditional views of niche specialization in methanogens, suggesting that *Methanotrichaceae* may act as a metabolic generalist under proper environmental conditions. Such flexibility allows for persistence across fluctuating redox environments, potentially explaining its dominance despite the various biochar treatments. This observation supports the idea that functional plasticity can coexist with stochastic assembly processes and provides evidence that taxa are not limited by deterministic niche constraints. Rather, they retain adaptive metabolic pathways to ensure ecological stability [88,89].

Our results indicate that CH_4_ was reduced without a corresponding reduction in CO_2_ levels, suggesting a decoupling of *Methanotrichaceae*’s abundance from its traditional methanogenic activity. This decoupling may be attributed to the shifts in electron flow dynamics within the microbiome. Biochar’s conductive properties can facilitate DIET, altering traditional syntrophic relationships and allowing for CH_4_ reduction regardless of CO_2_ levels remaining unchanged [90,91]. In addition, CO_2_ may have been redirected to alternative metabolic pathways such as anaplerotic carbon fixation or used as an electron sink by non-methanogenic microbial species [89]. These observations align with Vellend’s unified framework, where stochastic processes such as drift might allow for the persistence of taxa regardless of functional activity, while deterministic forces selectively influence metabolic outputs rather than taxonomic composition. The metabolic plasticity demonstrated here highlights the complexity of community assembly. Therefore, in accordance with previous work, our results suggest alignment with a neutral–deterministic continuum wherein functional redundancy and stochasticity coexist with niche-driven selection [79,81,85]. These results warrant further longitudinal investigation to fully untangle the assembly dynamics in swine manure anaerobic lagoons to identify key strengths and weaknesses of lagoon treatments such as biochar.

While there are many strengths to this study, there are several weaknesses that should be addressed in future investigations. First, this study was not a repeated-measures, longitudinal experiment. Larger sample sizes and longer sampling windows would allow for a more robust capture of the dynamic microbial ecology within anaerobic lagoons. While our study does provide insights into the nature of composition, diversity, and assembly of biochar-amended microbiomes, it is important to identify timescale changes to validate the findings within this study. Second, our study collected samples from one lagoon at a single swine operating facility. While our study design provides valuable baseline data that can inform future studies, a broader, multi-location study would be needed to validate the findings within the current study. These geographically larger studies could address the variation, if any, found within swine manure samples across different locations. However, previous research has demonstrated that swine manure wastewater samples and soil sediment samples were similar in composition across three different farms, although diversity varied [92]. Therefore, the findings within our study provide meaningful insights into anaerobic lagoon microbial ecology and GHG mitigation using biochar that could guide future research across various geographies. Third, certain important elemental observations were not included, primarily hydrogen sulfide (H_2_S). An increase in H_2_S is a primary mechanism for foul odors from anaerobic lagoons. While current research efforts have yet to elucidate the microbial communities’ influence on odor formation in swine manure anaerobic treatment systems [9], our results indicate that certain sulfur-reducing bacteria are enriched in biochar-amended systems. This finding is consistent with previous research showing that biochar can be effective in mitigating H_2_S emissions from manure [20,28]. However, the lack of direct tests with H_2_S or sulfur, in general, does not allow the current study to make valid claims on the microbiome’s importance regarding sulfur and odor. Fourth, evaluating stochasticity is challenging in empirical studies. As proposed by Evans et al. [93], stochasticity is often overestimated due to the mathematical calculations of within-group distance that accounts for variance due to error and unmeasured environmental factors. As such, designing studies with a more theoretical framework in mind could provide more robust investigations of microbial assembly dynamics [5,25]. Fifth, a true metagenomic approach could provide a better resolution of the mechanistic properties in biochar-amended microbiomes. While PICRUSt provides predictions on functionality and has been used in previous studies [22], it cannot provide genetic proof of these functionalities. Thus, a more comprehensive metagenomic approach could elucidate key genetic features that highlight the functionalities seen in these complex microbiomes.

## 5. Conclusions

Swine manure anaerobic lagoons are composed of complex microbial interactions. Proper lagoon management is crucial to utilizing these interactions for long-term storage and land-use applications. Biochar amendments provide a unique treatment opportunity for swine manure in anaerobic lagoons, considering their cost and ability to reduce GHGs and odorous emissions. The dynamics between biochar-amended swine manure and its microbiome could provide key insights into the mechanistic values of biochar as a lagoon treatment option. Herein, our pilot study demonstrated biochar’s ability to reduce CH_4_ despite microbial compositional shifts. The unified neutral theory-based assembly patterns demonstrated in this study are highlighted by the microbiome’s functional resilience. These results could provide researchers with a roadmap for future longitudinal investigations to elucidate the influence biochar has on microbial ecology on a temporal scale. Future research should utilize metagenomic shotgun sequencing to better describe the functional traits of the microbial communities. In doing so, researchers could gain better understanding into how functional redundancy and stochasticity enable swine manure anaerobic lagoon microbiomes to achieve functional resiliency and plasticity over time. As such, anaerobic lagoon treatment protocols using biochar could be used to engineer the microbiome for better long-term storage and land-use applications.

## Figures and Tables

**Figure 1 microorganisms-13-00758-f001:**
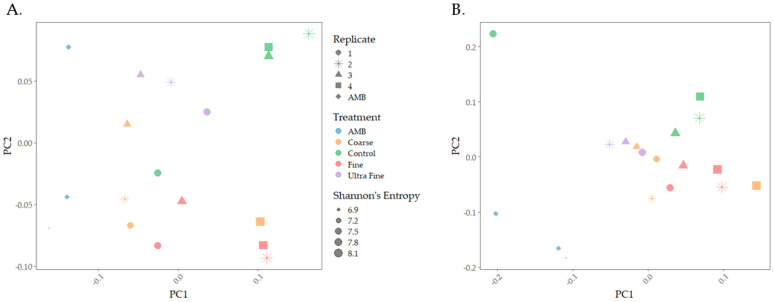
Microbial community composition PCoA plots. (**A**) Weighted UniFrac distance was used to analyze community composition. (**B**) Unweighted UniFrac distance was used to measure differences in community membership.

**Figure 2 microorganisms-13-00758-f002:**
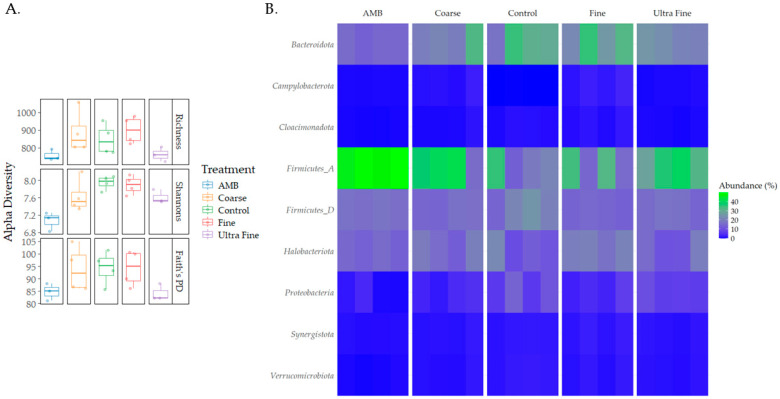
Diversity and taxonomic profiles. (**A**) Alpha diversity differences between treatment types were measured using observed richness, Shannon’s index, and Faith’s PD. (**B**) Taxonomic profiles for each treatment type were analyzed at the phylum level.

**Figure 3 microorganisms-13-00758-f003:**
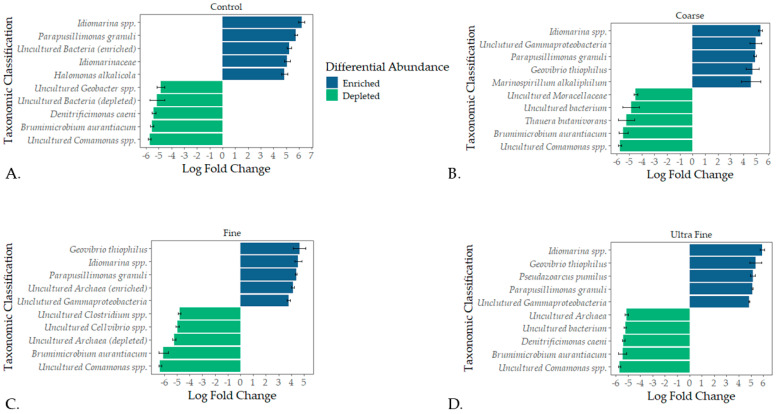
Differential abundance testing using ANCOM-BC. The top five enriched and depleted microbial taxa were recorded for CON (**A**), COR-biochar (**B**), FIN-biochar (**C**), and UF-biochar (**D**).

**Figure 4 microorganisms-13-00758-f004:**
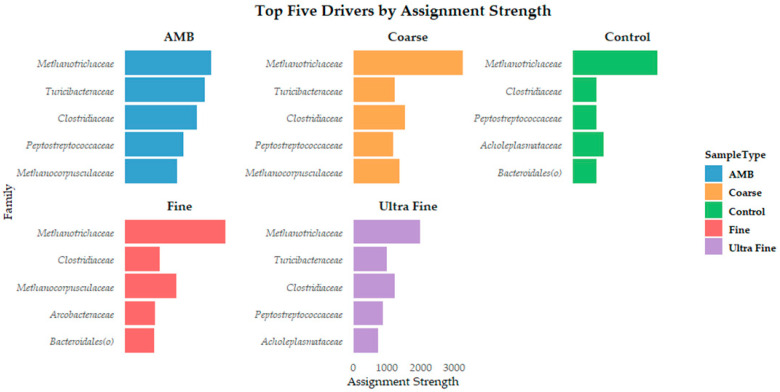
The top five microbial drivers by assignment strength at the family level for each treatment type.

**Figure 5 microorganisms-13-00758-f005:**
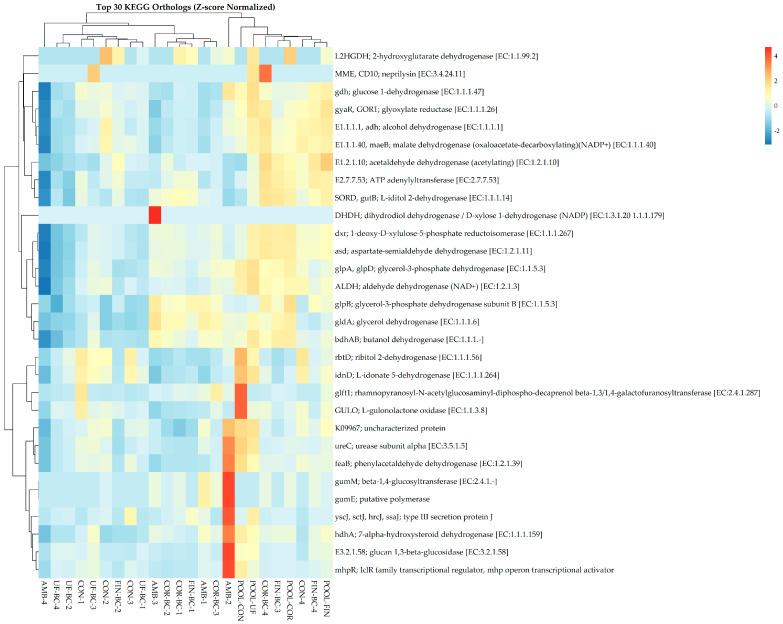
Dual dendrogram of the top 30 KEGG orthologs associated with each sample following Z-score normalization.

**Table 1 microorganisms-13-00758-t001:** Percent reductions in gaseous emissions by treatment type at day 14.

		Percentage Reduction (%R)			
Measurement	Biochar	Mean	Median	Standard Deviation	*p*-Value *
Ammonia (NH_3_)	Coarse	34.7	22.6	50.3	0.462
	Fine	45.4	43	13.1	
	Ultra-fine	35.9	51.9	44.2	
Methane (CH_4_)	Coarse	78.9	77.7	5.64	0.005
	Fine	88.2	86.3	5.14	
	Ultra-fine	93.2	93.7	3.8	
Water (H_2_O)	Coarse	19.8	19.4	9.37	0.040
	Fine	39.7	23.6	40.6	
	Ultra-fine	−0.206	7.36	35.8	
Carbon dioxide (CO_2_)	Coarse	−83.8	−75.9	23.7	0.033
	Fine	−107	−66.3	115	
	Ultra-fine	−79.2	−83.2	20.3	

* Kruskal–Wallis testing for statistically significant differences among treatment groups.

## Data Availability

The 16S rRNA gene sequencing data are publicly available in the NCBI SRA database under accession No. PRJNA1222416 (https://www.ncbi.nlm.nih.gov/bioproject/PRJNA1222416, accessed on 30 March 2025).

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
