# Peer review of "Characterization and Assembly Dynamics of the Microbiome Associated with Swine Anaerobic Lagoon Manure Treated with Biochar"

_microorganisms, 2025, doi:10.3390/microorganisms13040758_

Round 1

Reviewer 1 Report

Comments and Suggestions for Authors

The authors conducted a pilot study of the microbial community of a swine anaerobic lagoon. The authors, using the NGS approach and powerful statistical tools, showed that these lagoons have a complex microbial community. Thought, I have some major and minor remarks, in common the MS is suitable for publication in Microorganism journal.

Main concerns

Authors should significantly reduce the abstract, especially the parts where authors describe metagenomic methods and results. Abstract should be no more then 200 worlds.

In pairwise PERMANOVA analysis of weighted/unweighted UniFrac distance, the Authors postulated that significance difference is “p<0.05”. But for metagenomic studies, which include huge of data, the significance differences are when the “p” is below 0.001. In addition, the visual representation in Figure 1 shows that the graphs themselves have small values on the scale itself, and the values are not form distinct groups. The same question Also for other statistical analysis: “Shannon’s Index”. So, in discussion section Authors should speak about tendency, but not «significance difference».

The authors showed that microbial communities are not significantly different in differ experimental models.  But at the same time, gas emissions were strongly correlated with size of coal. So, Table 1 shows that the smaller the coal fraction possessed the less CH3 emissions. Doesn't it seem to the authors that it is not so much the microbial composition that influences, but the coal itself (I mean sorbtion property of coal )? I would suggest that the authors should discuss sorption property of coal in the Discussion section.

Minor concerns

Line 67: Replace “CH4.” To “methane (CH4)”

Line 67-68. Please, set reference to this sentence.

Line 135. For metagenomic research Authors should also use yet additional control sample – it is “Red Oak charcoal”. Although I understand that it is not possible to add such control to the already conducted research. But in their future studies I recommend that authors use additional control of “Red Oak charcoal”

Line 165-168. The authors used ASW and SILVA DB, but this is a very resource-intensive process. I think it would be useful for readers to know how much RAM and how many processors were used at least for "classifier" and "taxonomy", correspondingly. And it would be nice if the authors reported how much time it took. This is not critical information for MS, so I do not insist on including it in the article.

Line 168-169. After denoising, Authors should to use chimera-removing procedure. So in line 296, Authors wrought “chimera removal”

Line 169-170: “rarefied at a sampling depth of 60,000 ASVs”. The choice of depth for rarefaction should also be based on Rarefaction Curve Analysis. Please, could Authors explain the choose of 60,000 depth?

Line 170: “filter low-quality”. I don't understand the authors' use of low-quality in this context. In Line 162, Authors did “quality-filtered”. May be Authors mean reads frequency? And what about “low-level background noise” procedure? Please, clarify.

Line 172-173. Did Authors filter out “unassignment” reads?

Line 194 “p-value < 0.01”. Authors used 999 permutations, so the “Significant results” have to be below 0.001. Could Authors explant to the choose of “0.01” instead “0.001”?

Table 1. This table contains only three experimental group, but does not have control group. It would be logically if control group also would be included in the Table. Please, explain, why You did not include control group?

Line 305: “were likely”. In this sentence, the authors should still say exactly whether there are differences or not. I recommend that the authors rephrase this sentence.

Line 308-310. Authors declared that “PERMANOVA analysis indicated that there were statistical differences in” “community composition”. But for community composition, authors get p-value < 0.01. Still, statistical differences have to be below p>0.001.

Line 342-348. In these sentences, the authors describe the percentage of bacteria at the phylum level and the family level. It is not clear from the text, are these average values for the control and experimental samples, or max value for some of the samples? Please clarify.

In Line 468 and below, Authors discussed about sulfur-reducing

Author Response

Comments and Suggestions for Authors

The authors conducted a pilot study of the microbial community of a swine anaerobic lagoon. The authors, using the NGS approach and powerful statistical tools, showed that these lagoons have a complex microbial community. Thought, I have some major and minor remarks, in common the MS is suitable for publication in Microorganism journal.

First and foremost, the authors want to sincerely thank the efforts of reviewer one for their in-depth critique of the article. Without the expertise of the reviewer, the manuscript would not have been improved to its updated status. Reviewer one’s edits and suggestions helped to shape our manuscript into one we believe is adequate for publication. Therefore, we sincerely thank reviewer one for their time and commitment to this process. We know that the review process takes a bit of time, and we value the reviewer’s time. Thank you for your dedication and expertise!

Main concerns

Authors should significantly reduce the abstract, especially the parts where authors describe metagenomic methods and results. Abstract should be no more then 200 worlds.

The authors agree that the abstract was far too long. This was an oversight, and we apologize. The abstract has been reduced to 201 words.

In pairwise PERMANOVA analysis of weighted/unweighted UniFrac distance, the Authors postulated that significance difference is “p<0.05”. But for metagenomic studies, which include huge of data, the significance differences are when the “p” is below 0.001. In addition, the visual representation in Figure 1 shows that the graphs themselves have small values on the scale itself, and the values are not form distinct groups. The same question Also for other statistical analysis: “Shannon’s Index”. So, in discussion section Authors should speak about tendency, but not «significance difference».

The authors appreciate the reviewer’s thoughts on the significance level of 0.05. However, we believe that the alpha level is correctly set at 0.05 for multiple reasons. The first is regarding PERMANOVA. Anderson (2017; https://doi.org/10.1002/9781118445112.stat07841) describes in detail the statistical derivation of PERMANOVA and therein, details that alpha level 0.001 is the smallest possible p-value. There is no explicit statement that alpha level 0.001 must be used in these analyses. The p-value used in PERMANOVA is dependent on the number permutations used (i.e., 999). The higher number of permutations allows for smaller p-values. Secondly, Khomich et al. (2021; https://doi.org/10.1371/journal.pone.0259973) further compared alternative methods of multivariate statistical methods and set alpha level at 0.05 and 0.01. I encourage the reviewer to parse the QIIME2 forum as QIIME2 experts detail why setting the significance level at 0.05 is standard practice ( https://forum.qiime2.org/t/p-value-correction/12152). Additionally, several research articles have been published where significance levels 0.05, 0.01, and 0.001 are reported in microbiome studies. For example, Zhang et al. (2025; https://doi.org/10.1186/s40168-024-02010-9) analyzed Bifidobacterium in the hindgut of dairy calves and reported significance at 0.05, 0.01, and 0.001. For these reasons, the authors have chosen to leave the alpha level at 0.05. 

The authors showed that microbial communities are not significantly different in differ experimental models.  But at the same time, gas emissions were strongly correlated with size of coal. So, Table 1 shows that the smaller the coal fraction possessed the less CH3 emissions. Doesn't it seem to the authors that it is not so much the microbial composition that influences, but the coal itself (I mean sorption property of coal)? I would suggest that the authors should discuss sorption property of coal in the Discussion section.

The reviewer poses an insightful question. The sorption of the biochar could hold significant effects on ammonia and GHG emissions. For the scope of this study, the adsorption values were not assessed. These analyses are part of another sister study our group is currently preparing for peer review. Herein, we are not particularly interested in what the biochar is doing mechanistically and how its physichochemical properties (including pore sizes and sorption capacity associated with each of the biochar size fraction used) affect the emissions mitigation. Instead, we are interested in assessing how biochar impacts the microbiome and in turn, how the perturbed microbiome interacts with gaseous emissions.

Minor concerns

Line 67: Replace “CH4.” To “methane (CH4)”

Methane has been previously defined in line 42 of the revised manuscript. If it increases readability, the authors can make this change in the final rounds of revisions and finalizations! Please let us know!

Line 67-68. Please, set reference to this sentence.

Addressed!

Line 135. For metagenomic research Authors should also use yet additional control sample – it is “Red Oak charcoal”. Although I understand that it is not possible to add such control to the already conducted research. But in their future studies I recommend that authors use additional control of “Red Oak charcoal”

Thank you for this insight! In future studies, this will be addressed!

Line 165-168. The authors used ASW and SILVA DB, but this is a very resource-intensive process. I think it would be useful for readers to know how much RAM and how many processors were used at least for "classifier" and "taxonomy", correspondingly. And it would be nice if the authors reported how much time it took. This is not critical information for MS, so I do not insist on including it in the article.

The authors appreciate the reviewer’s comments on the processing time and power for the classifier steps. Each step used a total of ~40 cores on 1 priority node using the SCINet Ceres supercomputing cluster. Since this was done on a cluster, the processing time was significantly shorter than it would be if done on a local machine. I am unsure the RAM used in this process, but each classifying step takes roughly ~2-4 hours, depending on how many jobs have been submitted to the cluster prior to our scripts. Since this is usually not reported in microbiome focused studies, we have elected to keep the manuscript as is.

Line 168-169. After denoising, Authors should use chimera-removing procedure. So in line 296, Authors wrought “chimera removal”

Thank you for the comment. In our study, we utilized a commercial laboratory for sequencing and demultiplexing. The lab follows an in-house proprietary protocol for that step. We receive the demultiplexed sequences, which have been quality filtered for chimeras. We have placed a sentence here to qualify this in the methods (Line 151).

Line 169-170: “rarefied at a sampling depth of 60,000 ASVs”. The choice of depth for rarefaction should also be based on Rarefaction Curve Analysis. Please, could Authors explain the choose of 60,000 depth?

Thank you for this critique. In our analysis, while we did use the rarefaction curves, we also used the sampling depth tool within QIIME2 view. We have added comment on this in the Materials and Methods (Line 159-160).

Line 170: “filter low-quality”. I don't understand the authors' use of low-quality in this context. In Line 162, Authors did “quality-filtered”. May be Authors mean reads frequency? And what about “low-level background noise” procedure? Please, clarify.

We apologize for the lack of clarity. The sequences were quality filtered based on the Phred quality score. This has been corrected within the materials and methods (Line 161).

Line 172-173. Did Authors filter out “unassignment” reads?

No, we did not filter out unassigned reads. Filtering out unassigned reads introduces bias, especially if novel or poorly annotated taxa are present within the dataset. Additionally, in host-microbe systems, many sequences remain unclassified due to incomplete reference databases. These unassigned reads could represent novel taxa and removing them would remove their, if any, impact on microbial community.

Line 194 “p-value < 0.01”. Authors used 999 permutations, so the “Significant results” have to be below 0.001. Could Authors explant to the choose of “0.01” instead “0.001”?

This was addressed in a previous comment!

Table 1. This table contains only three experimental group, but does not have control group. It would be logically if control group also would be included in the Table. Please, explain, why You did not include control group?

This is an excellent question. Thank you! The control group is not included in the table because the %R is based on raw gaseous flux numbers. Since the control group results in 0 for the flux, %R is also 0 which inevitably produces NA values in statistical comparisons. For this reason, we chose to exclude the control from that table to give the reader a more streamlined representation of the %R between the treatment groups.

Line 305: “were likely”. In this sentence, the authors should still say exactly whether there are differences or not. I recommend that the authors rephrase this sentence.

Thank you! We have removed the phrase “were likely” from this sentence!

Line 308-310. Authors declared that “PERMANOVA analysis indicated that there were statistical differences in” “community composition”. But for community composition, authors get p-value < 0.01. Still, statistical differences have to be below p>0.001.

This has been addressed in a previous comment.

Line 342-348. In these sentences, the authors describe the percentage of bacteria at the phylum level and the family level. It is not clear from the text, are these average values for the control and experimental samples, or max value for some of the samples? Please clarify.

The percentages provided are for relative abundance. This has now been clarified in the results section (Line 334).

In Line 468 and below, Authors discussed about sulfur-reducing

This appears to be an incomplete comment! We look forward to learning more about the reviewer’s take on this part of the discussion.

Reviewer 2 Report

Comments and Suggestions for Authors

Microorganisms

Manuscript Draft

Manuscript Number: 3516148

Title: Characterization and assembly dynamics of the microbiome associated with swine anaerobic lagoon manure treated with biochar

Article Type: Research article

General Comments on MDPI Questions that Reviewers must answer:

  • Is the manuscript clear, relevant for the field and presented in a well-structured manner? 

This manuscript is written clearly, is very well-structured, and is potentially relevant to the field since it focuses on how biochar can reduce methane in anaerobic swine lagoons from a single commercial swine operation in the state of Oklahoma, USA, during the spring of 2024. Given the potential contribution of this research, this manuscript has potential but requires more improvement to warrant publication in MDPI Microorganisms. Please make the following SIX general improvements/clarifications:

1) In the last paragraph of the Introduction section, re-word the stated objective as the goal. Then re-phrase the three primary questions as stated objectives.

2) Please make clear in the writing that there is low/no variability in anaerobic swine manure samples. If there was variability, please explain this. Commercial swine facilities have enclosed manure pits under the hog facility which is not subject to outdoor environmental conditions. The feed is also fairly uniform. Please explain and justify what samples from only one facility during one month in one year is sufficient to rely upon for the study.

3) While the writing is OK, please improve the paragraph organization as some paragraphs are very long which makes the writing more difficult to understand. Paragraphs by definition have a minimum of 3 sentences (1 topic sentence followed by a minimum of 2 supporting sentences). Paragraphs should also flow from one to the next with adequate transitions. In the extremely long paragraphs, the topics of the writing change so these long tracts of writing need to be broken up into shorter, more digestible paragraphs.

4) There should be no abbreviations used in the Abstract nor figure/table titles nor section headings. Even though you provide an abbreviation guide at the end, I had to search the entire manuscript again to figure out what COR, FIN, and UF stand for. Please make it easy for the reader to understand without having to search around for the abbreviation guide. Please change BC to biochar throughout the manuscript since there is not much efficiency in space saved relative to ease of reading comprehension.

5) For Table 1, change to the heading of the 2nd column to Biochar and then in all rows below in that column change COR-BC to Coarse, FIN-BC to Fine, and UF-BC to Ultra-fine. Add a row on top then for the 3rd, 4th, and 5th columns write Percent Reduction in this row above these 3 columns with the side borders deleted so this is centered. Then change the row below for the 3 columns to Mean and Median and Standard Deviation. In the 1st column, change the writing to Ammonia (NH3), Methane (CH4), Water (H2O), and Carbon dioxide (CO2). There is a well-thought out art form for tables and figures as they should be “stand-alone” in understanding.

6) Please add a couple of sentences at the end of the Conclusion paragraph on how future research can expand upon the current work.

Please also make the following TEN minor edits and clarifications:

1) Change L1 to Article (what you have is the default format which is not correct)

2) Please capitalize all major words in the title.

3) The Abstract is too long since it is greater than 200 words.

4) On L38, the keywords need to be in alphabetical order with the first one capitalized with keywords separated by semi-colons (not commas): Keywords: Biochar; greenhouse gas emissions; microbial ecology; swine manure

5) The sub-headings should not have abbreviations (e.g., BC) so please write out.

6) Please add a blank row above and below equation 1.

7) Add blank row below Figure 2 caption below L359.

8) Delete the word Section on L364.

9) Add blank row below L381 and L404, etc. (there should be a blank row above and below every figure and table.

10) In the References, for citations that are NOT journal articles, the year should NOT be in bold (e.g., on L627, the year 2022 should not be in bold). Please correct this elsewhere.

  • Are the cited references mostly recent publications (within the last 5 years) and relevant? Does it include an excessive number of self-citations?

There are about 25 of the 92 cited references have been published within the last 4 to 5 years since 2020 and appear relevant to the research topic. There are no excessive self-citations.

  • Is the manuscript scientifically sound and is the experimental design appropriate to test the hypothesis?

The manuscript is scientifically sound and the experimental analyses are appropriate.

  • Are the manuscript’s results reproducible based on the details given in the methods section?

The manuscript’s experimental results are reproducible based on what is written in the 2. Materials and Methods section.

  • Are the figures/tables/images/schemes appropriate? Do they properly show the data? Are they easy to interpret and understand? Is the data interpreted appropriately and consistently throughout the manuscript? Please include details regarding the statistical analysis or data acquired from specific databases.

The quality of the figures are fine. Improvements need to be made to Table 1.

  • Are the conclusions consistent with the evidence and arguments presented?

The Conclusions are consistent with the evidence and arguments presented. Please add a few sentences at the end of the Conclusions section on how future research can improve on the current work.

  • Please evaluate the data availability statements to ensure it is adequate.

All Back Matter sections are fine.

Author Response

General Comments on MDPI Questions that Reviewers must answer:

  • Is the manuscript clear, relevant for the field and presented in a well-structured manner? 

This manuscript is written clearly, is very well-structured, and is potentially relevant to the field since it focuses on how biochar can reduce methane in anaerobic swine lagoons from a single commercial swine operation in the state of Oklahoma, USA, during the spring of 2024. Given the potential contribution of this research, this manuscript has potential but requires more improvement to warrant publication in MDPI Microorganisms. Please make the following SIX general improvements/clarifications:

First and foremost, the authors would like to thank the reviewer for their time and commitment to the revision of our manuscript. The reviewer’s expertise has helped shape the manuscript into a much better version. We understand the time commitment that the review process takes, and we are grateful to the reviewer for their critiques, comments, and recommendations.

1) In the last paragraph of the Introduction section, re-word the stated objective as the goal. Then re-phrase the three primary questions as stated objectives.

This has been addressed! Thank you!

2) Please make clear in the writing that there is low/no variability in anaerobic swine manure samples. If there was variability, please explain this. Commercial swine facilities have enclosed manure pits under the hog facility which is not subject to outdoor environmental conditions. The feed is also fairly uniform. Please explain and justify what samples from only one facility during one month in one year is sufficient to rely upon for the study.

The authors appreciate the reviewer’s critique of variability in the swine manure used in the study. Our study collected the manure from an open anaerobic lagoon that are typical in the US Eastern seaboard (e.g., North Carolina) and South (e.g., Texas and Oklahoma, where our stakeholders are located). The manure was not collected from enclosed manure pits underneath the hogs (deep pit manure storages capable of storing 1 year of manure are typical in the Midwest (e.g., Iowa). Because of this, the lagoon manure was exposed to environmental conditions sufficient to associate GHG emissions and changes to the microbial communities. We have added more clarification on where the manure as collected in line 108. We apologize for this oversight in the first version.

3) While the writing is OK, please improve the paragraph organization as some paragraphs are very long which makes the writing more difficult to understand. Paragraphs by definition have a minimum of 3 sentences (1 topic sentence followed by a minimum of 2 supporting sentences). Paragraphs should also flow from one to the next with adequate transitions. In the extremely long paragraphs, the topics of the writing change so these long tracts of writing need to be broken up into shorter, more digestible paragraphs.

We have addressed the lengthy paragraphs by breaking them down into smaller paragraphs. We hope this is to the reviewer’s satisfaction! Thank you!

4) There should be no abbreviations used in the Abstract nor figure/table titles nor section headings. Even though you provide an abbreviation guide at the end, I had to search the entire manuscript again to figure out what COR, FIN, and UF stand for. Please make it easy for the reader to understand without having to search around for the abbreviation guide. Please change BC to biochar throughout the manuscript since there is not much efficiency in space saved relative to ease of reading comprehension.

We have removed the abbreviations from the abstract. We have also shortened the abstract quite significantly as per the recommendations of the reviewers of this manuscript. We apologize for the oversight of leaving in abbreviations within the abstract. We have also changed BC to biochar throughout the manuscript and the supplementary material per the reviewer’s recommendations.

5) For Table 1, change to the heading of the 2nd column to Biochar and then in all rows below in that column change COR-BC to Coarse, FIN-BC to Fine, and UF-BC to Ultra-fine. Add a row on top then for the 3rd, 4th, and 5th columns write Percent Reduction in this row above these 3 columns with the side borders deleted so this is centered. Then change the row below for the 3 columns to Mean and Median and Standard Deviation. In the 1st column, change the writing to Ammonia (NH3), Methane (CH4), Water (H2O), and Carbon dioxide (CO2). There is a well-thought out art form for tables and figures as they should be “stand-alone” in understanding.

Table 1 has been formatted to align with the reviewer’s enhancements. We believe these changes are in accordance with the reviewer’s recommendations. Thank you!

6) Please add a couple of sentences at the end of the Conclusion paragraph on how future research can expand upon the current work.

We have added in a few sentences to address how future work can expand upon the current study’s findings (lines 587-591). Thank you!

Please also make the following TEN minor edits and clarifications:

1) Change L1 to Article (what you have is the default format which is not correct)

This has been addressed!

2) Please capitalize all major words in the title.

The title has been adjusted!

3) The Abstract is too long since it is greater than 200 words.

The abstract has been significantly shortened from ~300 words to 201 words.

4) On L38, the keywords need to be in alphabetical order with the first one capitalized with keywords separated by semi-colons (not commas): Keywords: Biochar; greenhouse gas emissions; microbial ecology; swine manure

The keywords have been corrected!

5) The sub-headings should not have abbreviations (e.g., BC) so please write out.

All sub-headings have been revised!

6) Please add a blank row above and below equation 1.

Addressed!

7) Add blank row below Figure 2 caption below L359.

Addressed!

8) Delete the word Section on L364.

Addressed!

9) Add blank row below L381 and L404, etc. (there should be a blank row above and below every figure and table.

Addressed!

10) In the References, for citations that are NOT journal articles, the year should NOT be in bold (e.g., on L627, the year 2022 should not be in bold). Please correct this elsewhere.

 Addressed!

  • Are the cited references mostly recent publications (within the last 5 years) and relevant? Does it include an excessive number of self-citations?

There are about 25 of the 92 cited references have been published within the last 4 to 5 years since 2020 and appear relevant to the research topic. There are no excessive self-citations.

Thank you!

  • Is the manuscript scientifically sound and is the experimental design appropriate to test the hypothesis?

The manuscript is scientifically sound and the experimental analyses are appropriate.

Thank you!

  • Are the manuscript’s results reproducible based on the details given in the methods section?

The manuscript’s experimental results are reproducible based on what is written in the 2. Materials and Methods section.

Thank you!

  • Are the figures/tables/images/schemes appropriate? Do they properly show the data? Are they easy to interpret and understand? Is the data interpreted appropriately and consistently throughout the manuscript? Please include details regarding the statistical analysis or data acquired from specific databases.

The quality of the figures are fine. Improvements need to be made to Table 1.

Thank you! We have addressed Table 1 according to the reviewer’s recommendations.

  • Are the conclusions consistent with the evidence and arguments presented?

The Conclusions are consistent with the evidence and arguments presented. Please add a few sentences at the end of the Conclusions section on how future research can improve on the current work.

Thank you! The conclusion has been adjusted accordingly!

  • Please evaluate the data availability statements to ensure it is adequate.

All Back Matter sections are fine.

Thank you!

Round 2

Reviewer 1 Report

Comments and Suggestions for Authors

I am pleased with the authors' answers. I would especially thank Authors for the clarification regarding "p" value. I viewed qiime forum and articles. I believe that this manuscript can be published in the journal in current form.

Author Response

I am pleased with the authors' answers. I would especially thank Authors for the clarification regarding "p" value. I viewed qiime forum and articles. I believe that this manuscript can be published in the journal in current form.

Thank you to the reviewer for their hard work on the revision process. Their edits made a significant impact on the quality of the paper! We appreciate the reviewer's time and effort!

Reviewer 2 Report

Comments and Suggestions for Authors

Microorganisms

Manuscript Draft

Manuscript Number: 3516148

Title: Characterization and assembly dynamics of the microbiome associated with swine anaerobic lagoon manure treated with biochar

Article Type: Research article

General Comments on MDPI Questions that Reviewers must answer:

  • Is the manuscript clear, relevant for the field and presented in a well-structured manner? 

This manuscript is written clearly, is very well-structured, and is potentially relevant to the field since it focuses on how biochar can reduce methane in anaerobic swine lagoons from a single commercial swine operation in the state of Oklahoma, USA, during the spring of 2024. Given the potential contribution of this research, this manuscript has potential but requires more improvement to warrant publication in MDPI Microorganisms. Please make the following THREE general improvements/clarifications:

1) In the last paragraph of the Introduction section, change L168 to “Therefore, the goal of this...”

2) It is not clear from the writing where the following was asked to be clarified. Please make clear in the writing what type of variability exists in the aerobic swine manure samples. In the Materials and Methods section, please explain and justify why samples from only one facility during one month in one year is sufficient to rely upon for the study. What confidence do you have that this one sample from one location (especially if this is open to variation in environmental conditions) and the results obtained allow you to make representative conclusions? This needs to be justified in response to reviewer and better justified in the manuscript in terms of writing. If there are issues related to how representative the results and implications of the research are then this needs to be explained in the Discussion section.

3) Thank you for correcting use of abbreviations. After L821, it is not clear how the following is biochar since there were only 3 types evaluated (ANCOM appears to be a method and not biochar):

ANCOM-biochar Analysis of composition of microbiomes with bias corrections

Author Response

General Comments on MDPI Questions that Reviewers must answer:

Is the manuscript clear, relevant for the field and presented in a well-structured manner?

This manuscript is written clearly, is very well-structured, and is potentially relevant to the field since it focuses on how biochar can reduce methane in anaerobic swine lagoons from a single commercial swine operation in the state of Oklahoma, USA, during the spring of 2024. Given the potential contribution of this research, this manuscript has potential but requires more improvement to warrant publication in MDPI Microorganisms. Please make the following THREE general improvements/clarifications:

The authors want to sincerely thank the reviewer for their continued insights and critiques. We value the reviewer’s time and efforts. Please find our responses below.

1) In the last paragraph of the Introduction section, change L168 to “Therefore, the goal of this...”

This has been addressed (line 100).

2) It is not clear from the writing where the following was asked to be clarified. Please make clear in the writing what type of variability exists in the aerobic swine manure samples. In the Materials and Methods section, please explain and justify why samples from only one facility during one month in one year is sufficient to rely upon for the study. What confidence do you have that this one sample from one location (especially if this is open to variation in environmental conditions) and the results obtained allow you to make representative conclusions? This needs to be justified in response to reviewer and better justified in the manuscript in terms of writing. If there are issues related to how representative the results and implications of the research are then this needs to be explained in the Discussion section.

The reviewer has made a valid and excellent critique. We apologize for not adequately addressing this comment in the previous round of revisions. Using a single location for microbial sampling is a common practice across many disciplines (i.e., BRD in cattle from one feedlot, https://doi.org/10.1186/s42523-022-00167-y; Listeria prevalence in a single meat-processing facility, https://doi.org/10.1128/spectrum.02045-22). This approach is valuable for controlling variation bias in intervention-based study designs. While we agree that a multi-location study could provide more robust findings, we are confident that the use of a single location provides valuable results that lay the groundwork for future, geographically larger studies. Furthermore, the current study collected manure samples from a swine finisher farm that is operated by a single company that has multiple farm locations, making it a representative industry sample. The farm management practices such as facility design, breeding and genetics, and feeding practices are therefore nearly identical across the farms. Thus, there would be in theory, minimal changes to microbial ecology. This is validated by previous research in ruminants in which various ruminant species and breeds across continental geographies share a common, core microbiome (https://doi.org/10.1038/srep14567). We have provided this context in the first paragraph of the materials and methods (lines 113-119).

Secondly, we agree that the reviewer’s concerns deserved more attention within the discussion. We have added several sentences discussing how single locations could be a limitation to the study (lines 572-581). Additionally, we have provided an additional reference that highlights that swine manure/wastewater microbiome compositions were similar across three separate farms (https://doi.org/10.1016/j.scitotenv.2019.05.369).

3) Thank you for correcting use of abbreviations. After L821, it is not clear how the following is biochar since there were only 3 types evaluated (ANCOM appears to be a method and not biochar):

ANCOM-biochar Analysis of composition of microbiomes with bias corrections

Thank you for pointing this out. We apologize for not making that change in the previous round of revisions. This has now been addressed (line 642, 3rd abbreviation).